# A System of Rapidly Detecting Escherichia Coli in Food Based on a Nanoprobe and Improved ATP Bioluminescence Technology

**DOI:** 10.3390/nano12142417

**Published:** 2022-07-14

**Authors:** Zhen Sun, Jia Guo, Wenbo Wan, Chunxing Wang

**Affiliations:** 1College of Physics and Electronic Science, Shandong Normal University, Jinan 250358, China; szskyedu@163.com (Z.S.); 17862967173@163.com (J.G.); 2College of Information Science and Engineering, Shandong Normal University, Jinan 250358, China; wanwenbo@sdnu.edu.cn

**Keywords:** nanoprobe, pathogen heat-treatment, adenosine triphosphate luminescence, graphene transparent electrode

## Abstract

Bacterial contamination is an important factor causing food security issues. Among the bacteria, *Escherichia coli* is one of the main pathogens of food-borne microorganisms. However, traditional bacterial detection approaches cannot meet the requirements of real-time and on-site detection. Thus, it is of great significance to develop a rapid and accurate detection of bacteria in food to ensure food safety and safeguard human health. The pathogen heat-treatment module was designed in this paper based on the techniques including nanoprobe, pathogen heat-treatment, graphene transparent electrode (GTE), and adenosine triphosphate (ATP) bioluminescence technology. The system mainly consists of two parts: one is the optical detection unit; the other is the data processing unit. And it can quickly and automatically detect the number of bacterial colonies in food such as milk etc. The system uses not only the probe to capture and enrich *E. coli* by antigen-antibody interaction but also the heat treatment to increase the amount of ATP released from bacterial cells within five minutes. To enhance the detecting accuracy and sensitivity, the electric field generated by GTE is adopted in the system to enrich ATP. Compared to the other conventional methods, the linear correlation coefficient of the system can be reached 0.975, and the system meets the design requirements. Under the optimal experimental conditions, the detection can be completed within 25 min, and the detectable concentration of bacteria is in the range of 3.1 × 10^1^–10^6^ CFU/mL. This system satisfies the demands of a fast and on-site inspection.

## 1. Introduction

In recent years, with the development of economic and social progress, human living standards have gradually improved, increasing attention has been paid to food safety, and requirements for food safety technology that are more stringent have been established. According to news reports and data, bacterial infections cause about 40% of all diseases and pose a huge threat to public health and the social economy [1]. Today, common bacterial detection approaches include culture and colony counting methods [2], enzyme-linked immunosorbent assay (ELISA) [3], biosensor technology [4], and adenosine triphosphate (ATP) and bioluminescence assay [5].

*Escherichia coli* (*E. coli*) is a common conditionally pathogenic bacterium. Under certain conditions, a part of *E. coli* strains harboring virulence factors are considered to be pathogenic to humans, so they are often used as an indicator for monitoring the quality of food and drinking water. Currently, the approaches to detecting *E.coli* have been divided into traditional detection methods and new modern detection methods. The traditional detection methods include the plate dilution method [6], membrane filtration, fluorescence quenching [7], and the use of quartz-crystal microbalance-based sensors [8]. Although these methods have relatively high reliability and sensitivity, most of them still have some drawbacks. The plate dilution method, for example, was once regarded as a classic microbial detection method. However, this approach is comparatively complex because it requires the culturing and enriching of bacteria in the lab prior to testing. Thus, the detection cycle is longer, making real-time detection impossible. It is difficult to meet the current domestic and foreign requirements of bacteria detection in food. New modern detection methods include ATP bioluminescence technology [9], magnetic-activated cell sorting (MACS) [10], the biosensor detection method [11], PCR detection technology [12], the gene chip method [13] and genome sequencing [14]. Among these modern detection methods, such as immunoassays, which are being applied in industry, most modern methods cannot meet the demand for portable and on-site detection due to the high equipment cost and professional laboratory operation. At the same time, most of the traditional bacteria detection methods generally have disadvantages, such as being time-consuming, complicated operation, low signal-to-noise ratio and non-specific bacterial identification, and these shortcomings lead that not all traditional methods can meet industry needs in every context. Therefore, it is very important to develop a fast, portable and highly accurate bacterial detection system to ensure food safety and maintain public health and environmental sanitation.

The combination of immunomagnetic separation technology and ATP bioluminescence technology provides a way to detect bacteria. However, how to improve the accuracy and expand the detection limit is an important question we face.

Immunomagnetic separation technology is one of the research hotspots in the field of bacterial detection. Magnetic nanoprobes are an indispensable part of immunomagnetic separation technology, based on nanotechnology and immunology, using various magnetic nanoparticles coated with immunoactive substances for immunological or biological analysis. Fish et al. realized a method to detect the concentration of *Bacillus spores* rapidly using immunomagnetic separation technology incorporating a chromatography technique in 2009 [15]. Moreover, Kuang et al. used immunomagnetic separation combined with a fluorescent probe to detect *Salmonella* with a sensitivity of 500 CFU/mL [16]. In 2016, Mengxu et al. designed an electrochemical immunosensor to detect *E. coli* and *Salmonella typhimurium* in food based on immunomagnetic separation technology and screen-printed interdigitated microelectrodes [17]. In recent years, immunomagnetic separation technology has developed rapidly and become a common detection method.

ATP bioluminescence technology is one of the most common microbiological detection methods, which can determine the total number of microorganisms in food. ATP is an important energy substance in cells that can store and provide energy. It is immanent in all cells and usually measured by the fluorescein–luciferase luminescence system. The wide adhibition of ATP bioluminescence technology in food detection and the medical industry is due to the creation and application of portable ATP Detectors. Murphy et al. used ATP bioluminescence technology to detect bacteria in liquid milk [18]. Although this method was proven feasible by comparing the results from the standard plate count method, it is difficult to use on-site. In 2017, Zhang et al. utilized ATP bioluminescence technology to detect the number of *E. coli* bacteria in food. The quantitative results demonstrated that their method was useful in detecting the bacteria with a sensitivity of 3.0 × 10^2^ CFU mL^−1^ [19]. Xu et al. combined transparent graphene electrodes with ATP bioluminescence technology to improve the accuracy of *E. coli* detection in food [20], but sensitivity remained low.

According to the research, temperature change has an obvious influence on the bacteria-detecting process. Heat-treatment pathogen technology is used to adopt heating, thermal insulation, and cooling to determine the properties or reaction states of pathogens under different temperatures, which can be better used in the field of bacterial detection or bacterial inactivation. Today, in addition to inactivating bacteria at high temperatures, the pathogen heat-treatment technology can be used to enhance the signal strength for bacterial detection. In 2017, Lee et al. found that the amount of ATP released from bacterial cells, such as *Salmonella enteritidis*, *E. coli O157:H7*, and *Bacillus cereus*, can be increased by subjecting the samples to heat treatment, which helps detect the number of bacteria colonies using ATP bioluminescence technology [21].

Because photoelectric conversion is applied in this system, graphene, a new bioassay material, has attracted our attention because of its excellent light transmission and electrical conductivity. In 2004, Novoselov et al. prepared graphene films by mechanical exfoliation and discovered their unique electronic properties [22]. Li et al. prepared graphene films by graphene chemical vapor deposition and developed a process to transfer graphene film to various substrates, making it possible to prepare transparent graphene electrodes [23]. In 2010, Bae et al. produced transparent graphene electrodes using a roll-to-roll production method and wet chemicals [24]. Afterwards, it was found that the optical transmittance of transparent graphene electrodes is 97.4% and that their conductance can be as low as 125 Ω^−1^, which is better than the other conventional transparent electrodes, such as those fabricated from indium tin oxide. Moreover, graphene also has excellent thermal conductivity, of up to 5300 W/m · K, higher than that of carbon nanotubes and diamonds.

Based on the above studies, our final goal is to further enhance the accuracy and sensitivity of the ATP bioluminescence detection system and expand its detection limit. Therefore, we designed a rapid detection system based on immunomagnetic separation technology, ATP bioluminescence technology, and pathogen heat-treatment technology. In addition, the system utilizes electric field force to enrich ATP. The system uses the biotinylated *E. coli* antibodies and streptavidin-modified magnetic nanoparticles to prepare the magnetic nanoprobes with an avidin-biotin link reaction. The detection tube is prepared by the chemical vapor deposition (CVD) of graphene [25]. The probes capture the pathogens, and then to enhance the amount of ATP the target pathogens release, heat treatment is carried out on the samples before the target pathogen is lysed. Among them, thin-film heaters and resistance temperature detectors (RTDs), the main components of heat treatment, have been widely applied in sensor chips, biological chips, and microfluid chips [26,27]. The released ATP can be enriched by an electric field applied through transparent graphene electrodes because ATP is negatively charged under weakly alkaline conditions. The experiment showed that the number of bacterial colonies is proportional to the luminescence intensity. As the intensity of ATP bioluminescence attenuates over time, we add another light source to reduce the error and use photomultiplier tubes (PMT) for luminescence acquisition and photoelectrical signal conversion. We then carry out a series of optimizations in the subsequent signal processing of the system to meet the requirements of measurement accuracy and achieve quantitative detection. Compared to the previous system, the heat treatment module is added, and the ATP bioluminescence reaction chamber is improved. Because temperature change has a significant impact on the bacteria-detecting process, the addition of heat treatment modules can eliminate the influence of ambient temperature change on ATP bioluminescence reaction, increase the relative luminescence unit (RLU), enhance the luminescence signal, and improve the accuracy and sensitivity of detection results. The results show that under the optimal experimental conditions, the linear correlation coefficient of the system could reach 0.975, and the detection concentration of bacteria was in the range of 1.7 × 10^1^–10^6^ CFU/mL. The combination of these technologies not only improves detection results and makes the system more convenient but also shows good performance in field detection.

## 2. Detection Principle

Adenine nucleoside triphosphate, abbreviated as adenosine triphosphate (ATP), is an energy-supplying cell substance. It exists in the cells of all kinds of organisms [28]. ATP consists of an adenine molecule, a ribose molecule and nucleotides formed with three linked phosphate groups [29]. ATP releases a large amount of energy when it hydrolyzes, the most direct energy source in living organisms. Figure 1 shows the molecular constitution of adenosine triphosphate.

In general, the quantity of ATP is roughly the same in each bacterium, at approximately 1–2 fg per cell [30]. Moreover, the mechanism, luciferin reacting with oxygen under the action of luciferase to produce bioluminescence, was discovered in 1947, and ATP can provide the energy for this reaction. According to our previous experiments, the experimental reagent’s luminescence intensity has a strong linear relationship with the concentration of ATP in a certain concentration range. The above findings provide theoretical bases for measuring the amount of bacteria utilizing ATP bioluminescence technology. In the experiments, we need to use a reagent group including an erase reagent, lysis reagent, luciferase, and fluorescein. The function of the erase reagent is to remove the effects of somatic cells and free ATP. The function of the lysis reagent is to lyse *E. coli* and release ATP. Luciferase and fluorescein provide raw materials for the bioluminescence process and react in the presence of Mg^2 +^ and O_2_. In addition, we subject the captured *E. coli* pathogen to heating during detection, which can increase the amount of ATP the pathogen releases and enhance detection sensitivity. In our previous study, we found that the reagent’s concentration affected the luminescence intensity. Therefore, we subjected the reagent concentration to 30–40 mg/L to stabilize the peak luminescence intensity. When the ATP concentration is less than 10^−7^ mol/L, we observe a strong linear relationship between the ATP content and light intensity when the other reactants are sufficient in the reaction process. Because the solution’s pH affects luciferase activity, it will directly cause the emitting light to deviate from its wavelength peak. The results of experimental data show that a stable peak wavelength of light is acquired when the pH is between 7.5 and 7.8. In this process, almost all of the energy from ATP is converted into light energy. The amount of *E. coli* cells can be measured indirectly using Equation (1), where hν represents luminous energy.
(1)$$\mathrm{ATP}+O_{2}+\mathrm{Luciferin}\overset{{\mathrm{Mg}}^{2+}+\mathrm{Luciferase}}{↔}\mathrm{Oxyluciferin}+\mathrm{AMP}+{\mathrm{CO}}_{2}+\mathrm{ppi}+h\nu \;$$

A nanoprobe is a nanoscale biosensor that can detect single living cells, which have the characteristics of nanoscale size and real-time monitoring and causes little damage to cells [31]. The immunomagnetic beads (IMB) separation technique is a new immunological technique that combines the high specificity of immunological response with the unique magnetic responsiveness of magnetic beads. It is a kind of immunological detection method with strong specificity and high sensitivity. IBMs are magnetic beads wrapped with monoclonal antibodies, which have the strengths of high speed, high efficiency and simple operation. Therefore, they have a wide range of applications in the enrichment and separation of bacteria [32]. Compared to the traditional methods, IBMs do not have the disadvantages of time-consuming techniques and a pre-enrichment requirement. Magnetic nanoparticles can be collected by a magnetic field, and this property can be used to combine bacteria with magnetic nanoparticles for specific separation and to capture target bacteria.

In our previous study, we designed medium-size IBMs (120–200 nm) to capture *E. coli* [19]. First, we utilized streptavidin to modify IMBs to obtain magnetic beads whose surface is modified with streptavidin. Then we mixed the *E. coli*-specific antibodies and biotin to get biotinylated *E. coli* antibodies. We prepared immunomagnetic nanoprobes by combining magnetic beads modified with streptavidin with biotinylated *E. coli* antibodies by enzyme-linked immunosorbent assay [33,34]. We used immunomagnetic nanoprobes to capture *E. coli* by antigen-antibody reaction, and we prepared the IMB/antibody–*Escherichia coli*-immune compounds. An antibody is an immunoglobulin produced by plasma cells differentiated from B cells in response to the stimulation of antigenic substances, and antibodies can specifically bind to the corresponding antigen. In our experiments, we studied *E. coli O157:H7* and used *E. coli O157:H7* monoclonal antibodies to capture them. These antibodies bound specifically to the somatic antigen (O antigen) and the flagellar antigen (H antigen) of *E. coli O157:H7*, respectively, to achieve the effect of capturing the bacterium. Figure 2 shows the preparation and capture processes. To enrich and isolate the magnetic immune compounds, we designed a magnetic field at the two polar ends of the samples.

Graphene is a material with a mono-layered two-dimensional hexagonal lattice structure [35]. Due to graphene’s excellent properties, such as good electrical conductivity [36], light transmission, excellent mechanical properties, and electron mobility at room temperature, graphene and its chemically modified derivatives can be used as graphene transparent electrodes. Moreover, graphene has high thermal conductivity, which allows us to apply it for heat and cooling.

In previous studies, we found that temperature has a significant impact on the amount of ATP bacterial cells release. Therefore, in this study, we found a method that improves the sensitivity by increasing the amount of ATP that *E. coli* releases by subjecting the samples to heat treatment [21]. Here, we designed a temperature control module accurately. A heat treatment unit comprises three parts: a thermal cycler, reaction chambers, and a temperature detection and signal feedback web. The thermal cycler comprises a heating component, a cooling component and a thermal signal controller. However, the great difficulties of designing the thermal cycle structure lie in the accurate control of temperature and excellent thermal uniformity around the reaction chamber. 

To solve those problems, we designed a thin-film heater. The thin-film heater, which is formed by depositing polycrystalline or metallic materials on the substrate surfaces [37,38], is very popular because of its advantages, such as low power consumption, high operation speed, and strong temperature control performance, and it is crucial for the portability of bacterial detection equipment. To meet space and environmental needs, we designed a flexible polyimide thin-film heater that mainly comprises external insulation material and internal heating wire. We chose polyimide as external insulation material because it has high-temperature resistance, high insulation and excellent mechanical tractility properties. Commonly, the materials of the heating wire used in thin-film heaters are divided into Fe-Cr-Al alloy and nickel-chromium alloy. Here, we selected nickel-chromium alloy as the material for the thin-film heater because it has excellent thermoelectric properties. It is difficult to deform at high temperatures, and it has the advantageous qualities of having a long service life, being non-magnetic, having strong corrosion resistance, etc.

A temperature detector is an essential part of the temperature detection and signal feedback unit. It is used not only to detect the temperature of the heater and the reaction chamber but also to feed the temperature signal to the MCU. The heating or cooling element is run by the thermal signal controller to maintain a stable temperature. We used the resistance temperature detector (RTD) in our system because of its high accuracy, high stability, and wide temperature-control range. In general, metallic materials are used for RTDs; temperature monitoring relies on the relationship between the resistance and the temperature [39,40,41]. Platinum, copper and nickel are commonly used to manufacture RTDs, for they have the advantages of a high-temperature coefficient and quick response to temperature changes. Moreover, they are easily manufactured into refined coils. Among them, platinum, with its high stability, does not react physically or chemically in a certain temperature range. Platinum-resistance temperature detectors have a wide measuring range. In addition, they not only have better linearity than thermocouples and thermistors, but they are also more accurate and stable temperature sensors than any we have obtained up to now.

## 3. Testing Prototype Design

The system primarily includes two parts. One is the optical detection unit, and the other is the photoelectric signal processing and display unit, as Figure 3 shows. 

The optical detection unit comprises four components. Reactor #1 is a capture and enrichment unit, reactor #2 is a bioluminescence reaction unit, and reactor #3 is a heat treatment module. Additionally, the unit contains an automatic filling-up unit.

Reactor #1 is the capture and enrichment unit, which is intended to use our designed immunomagnetic nanoprobes to capture *E. coli* pathogens and enrich bacteria by magnetic separation rack, as shown in Figure 3. The details of *E. coli O157: H7* cultured are as follows. *E. coli O157: H7* standard strains were added to the modified *E. coli* medium (37 °C, 24 h) and then homogenized in 300 μL sterile saline. After centrifugation at 5000 rpm, the precipitated supernatant was discarded and placed in saline suspension again. Finally, the original bacterial solution was obtained at a concentration of 10^7^–10^8^ CFU/mL. Multiple gradient concentrations of the samples were obtained by diluting the original bacterial solution. The samples were divided into two parts, one was for the conventional culture method, and the other was for detection by our method. In the following, we will describe the technical details. First, 50 µL of the immunomagnetic nanoprobes solution is added to the *E. coli* solution and wait 10 min for the probe to capture the bacteria. Due to utilizing an enzyme-linked immunosorbent assay to bind *E. coli* antibodies to the immunomagnetic nanoprobes, the probe can capture *E. coli* through antigen-antibody reaction. Second, after the *E. coli* is captured, magnetic separation is carried out. And under the action of the magnetic separation rack, the IMB/antibody–Escherichia coli-immune compounds are adsorbed onto the bottom of the test tube. Then, the supernatant is discarded after magnetic separation. Last but not least, to obtain the captured *E. coli* solution, the enriched samples are washed three times with 150 mL of cleaning solution (10-mmol L-1 PBS, 0.05% Tween-20).

Reactor #2 is the bioluminescence reaction unit, which consists of a special detection tube and a photomultiplier (PMT). The detection tube is characterized by a graphite layer attached to the bottom. The heat treatment module (Reactor #3) is a temperature control device implemented using a heating component, a cooling component and a thermal signal controller. They cooperate to process the captured *E. coli* solution. First, the solution used for the ATP bioluminescence testing is subjected to heat treatment to increase the amount of ATP released from *E. coli*. We designed a kind of flexible thin-film heater as the heating component in the temperature control device, and its design was as shown in Figure 4. We used the polyimide film as the substrate material (thickness 0.15 mm). Nickel-chromium alloy foil was selected as the resistance circuit material (thickness 0.05 mm). Nickel-chromium alloy foil was laminated onto the substrate (polyimide film). This step was achieved using a thin thermosetting adhesive layer with excellent bonding properties when bonded to both materials. After lamination was stabilized, the substrate was drilled and positioned so that each layer within the heater remained aligned during a period of manufacturing. Next, the photoresist was smeared evenly on the panel (nickel-chromium alloy foil). After the photoresist was attached well, we placed the advanced photomask tool on the resist. The mask, which was generated based on a CAD design tool, was the final design template of the heating element, including the applicable element width required to produce the correct resistive heater. Then, the exposure was carried out. In this process, the resist was exposed to ultraviolet light to cure for use as a chemical etching resist. Next, the uncured resist was removed to expose the foil for etching and foil removal. At the same time, the cured resist can protect the pattern of the heater element. The panel was then chemically etched, stripped, and cleaned to chemically remove the foil not protected by the resistor and to retain the heating element pattern on the panel. The next step was to place a top overlay on the panel and laminate it. The top layer, the polyimide film (thickness 0.15 mm), provided access to the heating elements, including connecting wires and the openings for assemblies. Finally, after the heating film was removed from the panel, the wire was connected to the heating film, and the high-performance pressure-sensitive adhesive was coated on the back of the heating film. We adhered the heating film to the inner wall of the container that holds the test tube, and the heating film was wrapped tightly on the outer wall of the test tube for heating. Resistance temperature detector (RTD) PT100′s connector was fixated in the reserved position between the thin-film heater and the test tube for monitoring and feedback on the temperature of the test tube. Both the thin-film heater and RTD were controlled by the external drive circuit and the thermal signal controller. The thin-film heater was controlled using proportional-integral-derivative (PID) closed-loop controllers, which were implemented on CPU-ADUC834. CPU-ADUC834 is a microcontroller that has two independent ADCs [42]. In addition, the thin film heater was driven by a double MOS parallel active output circuit. A PT100 resistance temperature detector (RTD) was connected with a MAX31865 chip to convert the voltage signal corresponding to test tube temperature into a digital signal. The MAX31865 chip was then connected with an ADUC834 to control the drive circuit according to the collected temperature. Because the optimal temperature for the bioluminescence reaction is 24.5 °C, we heated the tube to 50 °C for 5 min and then let it cool to 24.5 °C for 20 s for the next bioluminescence reaction. The cooling component was implemented by a low-power fan, which was also controlled by the ADUC834.

ATP bioluminescence reaction occurred in the detection tube in the fluorescent reaction unit (Reactor #2). We connected the graphene electrode to the upper electrode through a steady current source, creating a weak electric field in the detection tube. The studies show that graphene has good optical transmittance, so the PMT, which is used for light trapping and photoelectric conversion, can be placed on the side of the test tube. To avoid the interference of external light as much as possible to ensure the accuracy of detection, the entire bioluminescence reaction and light capture and detection by PMT were performed in a completely closed environment. The procedure for this part of the assay is as follows: 30 μL of enriched *E. coli* suspension was added to the special test tube with a graphene electrode attached to the bottom. The solution in the tube is heat-treated for 5 min using the flexible thin-film heater we designed and then cooled to the optimum ATP bioluminescence reaction temperature using the cooling fan. Then 270 μL of the detection reagent was accurately injected with a peristaltic pump. The lysis agent in the detection reagent can lyse *E. coli* and release ATP. Meanwhile, the graphene electrode is energized by a built-in current source to create an electric field outside the test tube. Moreover, the experiments indicate that the luminescence from the ATP bioluminescence reaction is stronger under weak alkaline conditions compared to other acid-base conditions, and ATP has a negative charge under weak alkaline conditions. ATP will be enriched at the bottom of the test tube under the action of the electric field. The electric field also can enhance the activity of ATPASE and accelerate the oxidation of luciferase, thus enhancing the stability and accuracy of the system. After adding the detection reagent, the energy for the bioluminescence reaction is provided by ATP, and the luminescence emitted from the ATP bioluminescence reaction is captured by the PMT. The system automatically reads the optical signal after 60 s. And an internal calibration and compensation unit was used to detect and compensate for light attenuation, which will be described in the following. It is essential to add a filtering circuit in the design to reduce external noise interference due to the unavoidable system noise and weak input signals.

For portability and automation, we added an automated filling-up unit to the system, which is controlled by a microcontroller and enables the system to automatically select the appropriate reagents and quickly fill the required dose of reagents as required, as shown in Figure 5. In this unit, a single reagent hose line is represented by a thin line, and multiple reagent line hoses are represented by thick lines. For the tubes used in each unit, we choose the hoses characterized by high elasticity, low adhesiveness and low permeability. The different types of reagents for system selection are stored in different containers. Erase reagent, lysis reagent (50 mmol/LTris-HCl, 150 mmol/L NaCl, 0.02% NaN_3_, 100 μg/mL PMSF, 1 μg/mL Aprotinin, 1% Triton X-100) and bioluminescent reaction reagent (luciferase/luciferin reagent substrate, from Beijing YPH Biotechnology Co., Ltd., Beijing, China) is stored in three tubes a, b and c, respectively. Using a peristaltic pump in the automatic filling device, the required reagents are added to each unit via hoses to complete the test.

Although the test is carried out in a closed device, some external light still enters the reactor through cracks and transparent tube refraction, which forms scatting light. The light emitted by ATP luminescence is also absorbed by the solution because the ATP luminescence reaction takes place in the solution. These two points may affect the testing results, so the optical calibration method is adopted for compensation, and its schematic diagram is shown in Figure 6. Figure 6a is the schematic diagram without using the optical calibration method. The sample and detection reagent react to release bioluminescence in an optical reaction cell, which is directly detected by photomultiplier tubes. Therefore, it is inevitable to be affected by the absorption of luminescence and scatting light. To solve this problem, a light-emitting diode (LED) is installed on the opposite side of the photomultiplier tube, as shown in Figure 6b. The luminescence wavelength of the LED is required to be consistent with ATP bioluminescence, whose peak wavelength is 562 nm. And under the control of a constant current source, make its luminescence as stable as possible. When no detection reagent is added, the photomultiplier tube first detects and records the original light intensity of the LED and then turns it off. The approaches to close include power supply and shutter. After adding the detection reagent and bioluminescence detection, the LED is turned on again, and its luminescence is detected by a photomultiplier tube after passing through the optical reaction cell. Because the absorption and scattering of the light attenuate it, the detected bioluminescence signal can be corrected by the degree of attenuation to achieve the calibration function. 

After processing, the bioluminescence reaction is carried out in a special detector tube in reactor #2, with graphene electrodes attached to the bottom. The tube is chosen to use a high-transmittance glass test tube to improve the light transmission. The PMT is responsible for collecting the light signal, obtaining the relative luminous intensity, and completing the photoelectric conversion. As shown in Figure 7, current signals derived from photoelectric conversion are turned into analog voltage signals by an integrated operational amplifier circuit and become digital signals after analog-to-digital conversion. Then they are sent to a microcontroller for processing. A complete inspection process is controlled in 30 min. Moreover, to automate and improve the system’s efficiency, we designed an automatic cleaning function, which can be automatically cleaned between two inspection processes. The waste liquid obtained after cleaning is delivered to the waste liquid pool. This function can reduce the interference between two consecutive assays.

## 4. Results and Discussion

### 4.1. System Noise and Signal Level

During the testing process, the accuracy of the measurement results is improved using a filtering circuit in signal processing due to the interference from system noise and weak input signals are two important factors affecting the measurement accuracy. The specific filtering process is as follows. Firstly, the signal from PMT acquisition and photoelectric conversion is sampled and integrated. The sampling integral processing consists of two processes: sampling and integration. The sampling interval is determined according to the accuracy of the signal to be recovered. Then the sampling is integrated, and the integration process is implemented with an analog circuit. Both sampling and integration can be implemented using an integrating sampler, whose purpose is to extract the weak input signal from the noise. Later, the obtained samples are accumulated synchronously using an RC low-pass filter to suppress the noise and extract the desired signal, which takes advantage of the non-correlation between the noise and the signal. Figure 8 shows the system noise and the effective signal levels before and after filtering. It is clear from the line chart in Figure 8 that the effective signal levels before filtering ranges from 13 mV to 17.5 mV, and the effective signal levels after filtering ranges from 14.5 mV to 16 mV, which has a significant difference between the two sets of results. The system noise level ranges from −0.2 mV to 0.15 mV. Thus, it can be observed that the output signal level is smoother, and the output signal results are more stable than before filtering.

### 4.2. The Efficiency of Binding E. coli Antibodies to Immunomagnetic Beads

For the immunomagnetic nanoprobes to capture *E. coli*, the IMBs are required to bind with a sufficient amount of *E. coli* antibodies. In this study, the binding efficiency refers to the binding efficiency of the IMBs to the antibodies. Many instruments are available to measure the binding efficiency, and we chose the UV spectrophotometer. The specific operation is as follows: The absorbance of the *E. coli* antibody solution before and after adding IMBs is measured separately with the UV spectrophotometer. When IMBs are not yet bound to the antibody, the absorbance of the solution before the addition of IMBs can be recorded as A0. When the binding of IMB and antibody is completed, the absorbance of the solution after the addition of IMBs can be recorded as A1. The binding efficiency (R_C_) can be calculated using Equation (2).
(2)$$\mathrm{Rc}=\left[\frac{A0-A1}{A0}\right]\times 100\%\;$$

The results obtained from the preliminary data analysis in Table 1 show the average binding efficiency of 86.3%, with a maximum fluctuation of 1.4%. The above results indicate that the binding efficiency of IMB and *E. coli* antibodies is satisfactory, which meets the requirements of the binding efficiency of the assay. Moreover, according to the results of the subsequent experiments, the efficiency of immunomagnetic nanoprobes for capturing *E. coli* reaches 99.5%, which can meet the detection requirements for capturing accuracy.

### 4.3. ATP Bioluminescence Detection

During the detection process, the intensity of the luminescence signal gradually decayed over time due to the consumption of the substrate. To calculate the concentration of *E. coli* in solution using the luminescence intensity at the initial moment, we need to record the luminescence intensity at the initial moment. Therefore, we should perform ATP bioluminescence reaction kinetic fitting experiments. The experimental method is as follows. We choose standard ATP solutions with five different concentrations of 10^−15^, 10^−14^, 10^−13^, 10^−12^ and 10^−11^ mol/L. Under the same circumstances, different concentrations of standard ATP solutions and sufficient amounts of other reactants are reacted, which is repeated five times. The average of the five testing results is taken as the final experimental data, and the results are shown in Figure 9. It is apparent from Figure 9 that the luminescence intensity of ATP is negatively correlated with time, and the higher the concentration of ATP at the same experimental moment, the stronger the luminescence intensity of the solution. The experimental data of the samples with a concentration of 10^−13^ mol/L can be compared with the other data in Figure 9, which shows that the initial luminescence intensity produced by the samples with the concentration of 10^−15^ and 10^−14^ mol/L are too low and easily disturbed by noise. The luminescence intensity produced by the standard ATP solutions of 10^−12^ and 10^−11^ mol/L is not detected in the first 100 s because its intensity is too high beyond the detection range. Therefore, to improve the accuracy of the detection results, we choose samples with a concentration of 10^−13^ mol/L to perform the kinetic fitting of the ATP bioluminescence reaction. 

Figure 10 indicates the bioluminescence reaction kinetic fitted curve of the sample with a concentration of 10^−13^ mol/L. A closer inspection of Figure 10 shows that the output voltage obtained from the photoelectric conversion decays from an initial value of 2187.63 mV to 1013.72 mV within the first 100 s. The data in this figure are consistent with the kinetic Equation (3) [43], which will be used in subsequent experiments to calculate the initial luminous intensity:(3)$$y=A1\times {\;e}^{-\mathrm{Kt}}+y0\;$$
where *y* is the output voltage corresponding to the luminous intensity at time t, A1 is the output voltage corresponding to the luminous intensity at the initial moment, K is the attenuation constant, and y0 is the error. The calculated correlation coefficient of the fitted curve is 0.9943.

### 4.4. Effect of pH on Test Results

Luciferase is indispensable for ATP bioluminescence reaction, whose activity is easily affected by pH value. The experiments indicate that the fluorescence from ATP bioluminescence reaction is stronger under weak alkaline conditions than in other acid-base conditions. To achieve the desired effect of the bioluminescence reaction, we need to gain the pH interval where the luciferase activity reaches its peak. To obtain this pH interval, ATP solutions with a concentration of 10^−13^ mol/L were used for the test. The luminescence intensity of the solution, which reflects the luciferase activity, could be measured by a fluorescence detector. The details of the experiment are as follows. Multiple experiments were performed with the same concentration of ATP solution, and the pH value of the solution was adjusted with HCL and NAOH. Experiments showed that the more suitable temperature for ATP bioluminescence reaction is 24.5 °C [19], so the temperature of this experiment was controlled within a range of 24 ± 1 °C. The experimental results are shown in Figure 11. It is apparent from Figure 11 that the luminescence intensity is almost 0 when the pH value is too large (greater than 10) or too small (less than 4), which may be that the luciferase is inactivated and the reaction cannot be carried out. The luminescence intensity is significantly better when the pH value is between 7.5 and 7.9, and the luminescence intensity reaches its peak when the pH value is 7.7, which is 31.6 × 10^4^ RLU. Thus, the pH value of the solution in detection should be adjusted to a weak alkaline condition. ATP has a negative charge under weak alkaline conditions, which is conducive to the enrichment of ATP by the electric field force.

### 4.5. System Precision

System precision is a prerequisite to ensuring the system’s accuracy, which can be expressed by the reproducibility of the measurement results. Although high precision cannot guarantee high accuracy, low precision must not have good accuracy. Precision refers to the degree of dispersion between the measured data in the measurement of multiple identical samples. Coefficient of variation (*CV*) is generally used to determine the level of precision. *CV* can be calculated by Formula (4):(4)$$CV=\frac{SD}{Mean}\times 100\%\;$$
where *SD* is standard deviation; *Mean* is the average value. A low coefficient of variation demonstrates good testing precision, while a high variation coefficient indicates poor testing precision. We used 10 independent bacterial sample solutions with a concentration of 3 × 10^5^ CFU/mL to analyze the system precision in the experiment, and the results are shown in Figure 12. A simple analysis of the data in Figure 12 showed that the coefficient of variation was 3.96%.

### 4.6. Linear Relationship

In general, the lower limit of detection (LLD) is defined as the point where the signal-to-noise ratio equals three [44]. In this study, we applied the standard deviation of luminescence intensity as the system noise and calculated the lower limit of detection for luminescence intensity as 600 RLU. We selected several standard *E. coli* solutions ranging from 10^1^ to 10^7^ CFU/mL as sample solutions to test the relationship between luminescence intensity and bacterial concentration. The tests were divided into two groups: solutions without heat treatment and solutions after heat treatment. Figure 13 demonstrated that the luminous intensity was linearly related to the *E. coli* concentration. The linear correlation coefficient was 0.972 before and 0.975 after heat treatment. For the solutions without heat treatment, the luminescence intensity was below the lower detection limit when the concentration of *E. coli* was below 10^2^ CFU/mL. For the solutions after heat treatment, the system cannot detect when *E. coli* concentration is below 30 CFU/mL. As Figure 13 shows, there was a significant difference between the two groups of fitted curves. Comparing the fitted curves before and after heat treatment, it can be seen that the RLU of the heat-treated solution increased several times in the solution containing the same concentration of *E. coli.* Thus, the limit of detection (LOD) of *E. coli* detection using the ATP bioluminescence technique was increased by about one magnitude because of the enhanced bioluminescence signal. Our results suggest that the heat treatment step of the pathogen is useful for improving the sensitivity of the ATP bioluminescence technique for the detection of *E. coli*. As Table 2 shows, our results suggest that adding a pathogen heat-treatment step is useful to enhance the sensitivity for *E. coli* detection by the ATP bioluminescence technique.

### 4.7. Detection Accuracy

Absolute error is often regarded as a classical parameter to measure detection accuracy, which can be described as:(5)E=ABS[log(PM)−log(TCM)]
where *E* is the absolute error, *PM* is the detection results of the present method, and *TCM* is the detection results of the traditional culture method. Generally speaking, we regard the traditional culture method results as actual values and the present method results as measured values. When E < 1, the present method results and the traditional culture method results can be considered almost identical [48]. The detection system tested various foods and beverages, and the results were compared with those of the traditional culture method to calculate absolute error, represented in Figure 14b. Samples included four categories: drinks, meats, grain and quick-frozen food, Examples of items included milk, juice, beef, and frozen fish from supermarkets and farmers’ markets. To prepare the testing samples, we used the dilution method and the mixture ratio method [20]. In processing the experiment, the experimental conditions were adopted as follows: ATP bioluminescence reaction was preceded by heat treatment (50 °C, 5 min), and then the sample solution was cooled to the optimum temperature for bioluminescence reaction (24.5 °C, pH 7.4). During the assay, for solid and semi-solid samples, such as beef, we mixed 25 g samples with 225 mL normal saline in a sterile homogenization cup and homogenized it at 5000 rpm for 3 min to produce a 1:10 dilution solution. Liquid samples, such as milk, were collected with a sterile pipette. 25 mL sample was mixed with 225 mL normal saline in a sterile conical flask, and then it was shaken well at 200 rpm for 3 min to produce a 1:10 homogenized sample. Figure 14b shows the testing results before the addition of the heat treatment module, and Figure 14d shows the testing results after the addition of the heat treatment module. The comparison shows that the detection accuracy was more than 94% after adding the heat treatment module, especially in beverage and grain samples. The detection accuracy of quick-frozen food was also significantly improved.

## 5. Conclusions

*Escherichia coli* is a conditionally pathogenic bacterium, and some of its strains carrying virulence factors are pathogenic to humans. *E. coli* detection research has been a hot issue in the food hygiene and health field. This study is undertaken to design a real-time, rapid and accurate portable *E. coli* detection system and evaluate the detection effectiveness of the system. The system is innovative in combining three technologies: immunomagnetic separation technology, improved ATP bioluminescence technology, and graphene transparent electrodes. Among these, the improved ATP bioluminescence technology is a combination of heat-treatment pathogen technology and ATP bioluminescence detection to further improve detection range and detection accuracy. Based on the analysis of the testing results, the system can be used to detect *E. coli* in food and beverages with a detection accuracy of more than 94%. The detection time was within 30 min, and the range of detection colonies was 3.1 × 10^1^ to 10^6^ CFU/mL. The coefficient of variation was 3.96%, indicating that the system has high reliability and reproducibility. In addition, the system’s results showed a good linear correlation with those obtained by the traditional detection method, with a linearity coefficient of 0.975. Compared with the unimproved system [18], the detection accuracy was higher, the detection range was larger, the coefficient of variation was lower, and the linear correlation was stronger. A limitation of this study is that the system can only detect *E. coli* and cannot detect multiple pathogens simultaneously. In the future, further studies will improve on this limitation. 

## Figures and Tables

**Figure 1 nanomaterials-12-02417-f001:**
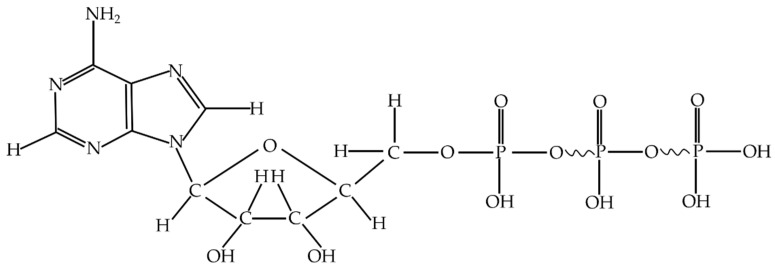
Structure of Adenosine triphosphate.

**Figure 2 nanomaterials-12-02417-f002:**
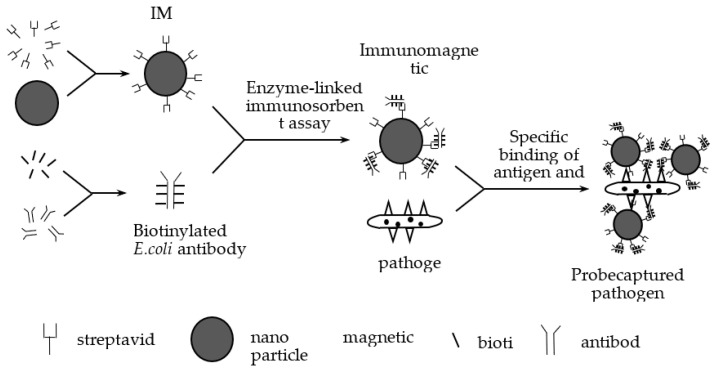
Magnetic nanoprobe and capture of *E. coli.*

**Figure 3 nanomaterials-12-02417-f003:**
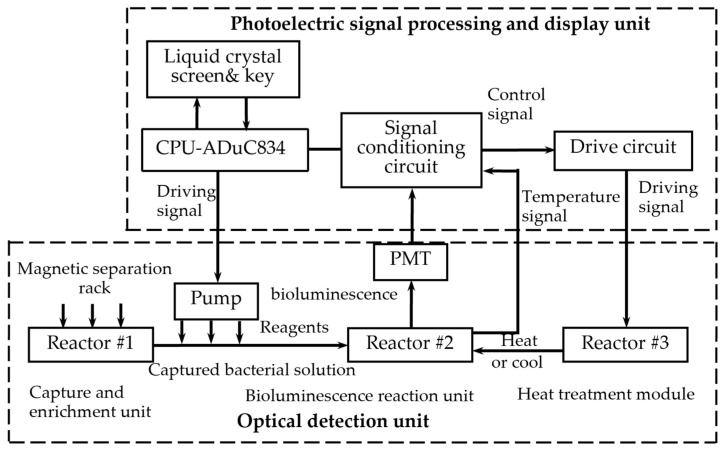
Frame diagram of the system. Reactor #1 is the capture and enrichment unit, which is intended to capture *E. coli* pathogens by our designed immunomagnetic nanoprobes and to enrich bacteria by a magnetic separation rack. Reactor #2 is the bioluminescence reaction unit, and reactor #3 is the heat treatment module, both of which cooperate to process the captured *E. coli* solution by pathogens heat-treating and adding reagents by a peristaltic pump. The bioluminescence signal is collected by the PMT and sent to the CPU through the signal processing circuit. The control signals of the peristaltic pump and the heat treatment module are sent by the CPU. The liquid crystal screen displays the results, and the device can be operated by key.

**Figure 4 nanomaterials-12-02417-f004:**
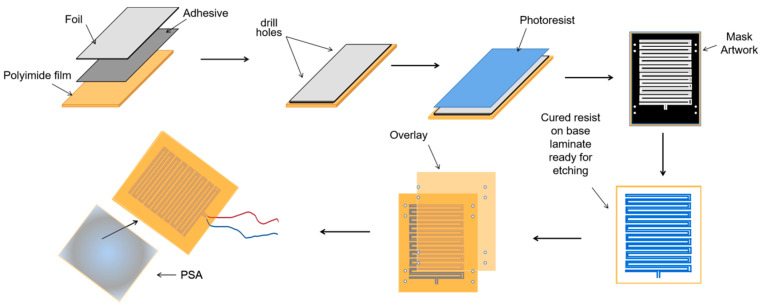
The production flow chart of the thin-film heater.

**Figure 5 nanomaterials-12-02417-f005:**
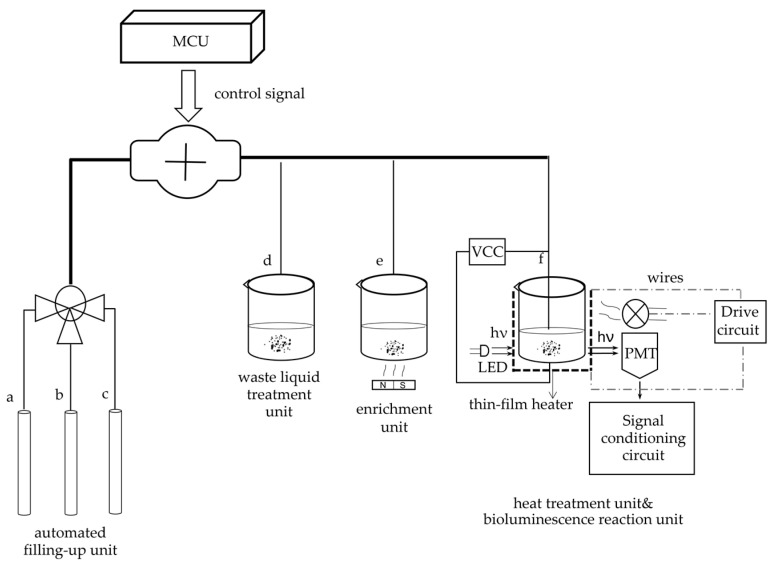
Automatic filling-up unit. *Escherichia coli* is captured using immunomagnetic nanoprobes and enriched in tube e. Pathogen heat treatment and bioluminescence reaction are performed in tube f. Tubes a, b and c contain erase reagent, lysis reagent and bioluminescence reagent, respectively. The peristaltic pump is controlled by MCU to fill the required reagents during the detection process. Tube d is used to store the waste liquid.

**Figure 6 nanomaterials-12-02417-f006:**
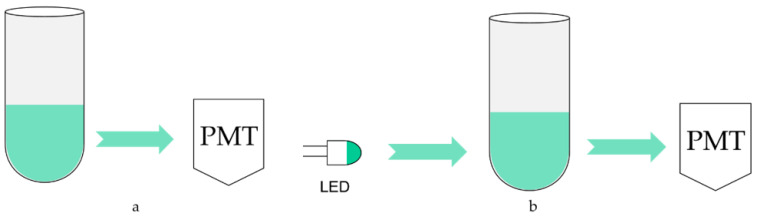
The schematic diagram of the optical calibration method. (**a**) PMT luminescence collection without optical calibration; (**b**) PMT luminescence collection with optical calibration.

**Figure 7 nanomaterials-12-02417-f007:**
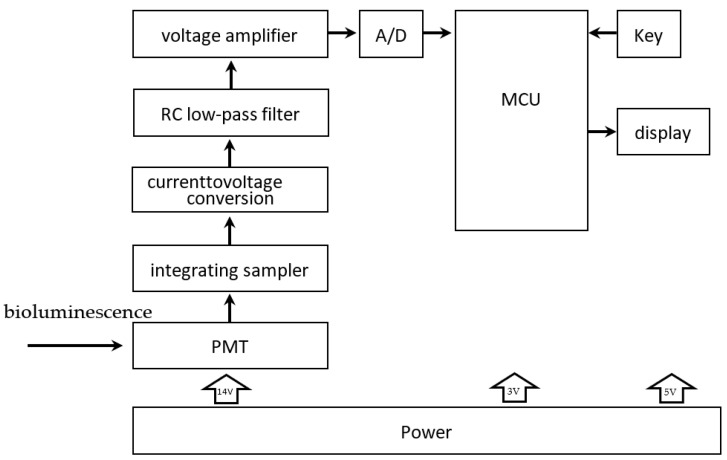
Signal transmission unit. The bioluminescence signal is converted into the current signal by the PMT. The current signal is sampled, and current-to-voltage is converted and then sent to the RC low-pass filter for filtering. The filtered analog voltage signal is amplified by the voltage amplifier circuit, which becomes a digital signal after analog-to-digital conversion and is transmitted to the MCU.

**Figure 8 nanomaterials-12-02417-f008:**
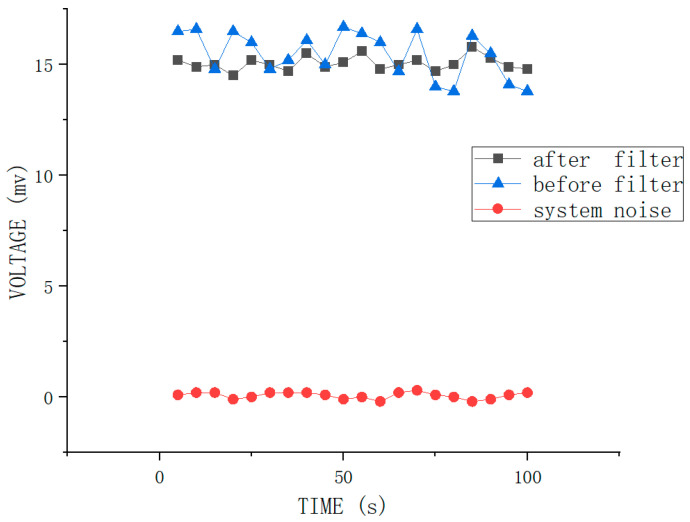
System noise and output signal level before and after filtering.

**Figure 9 nanomaterials-12-02417-f009:**
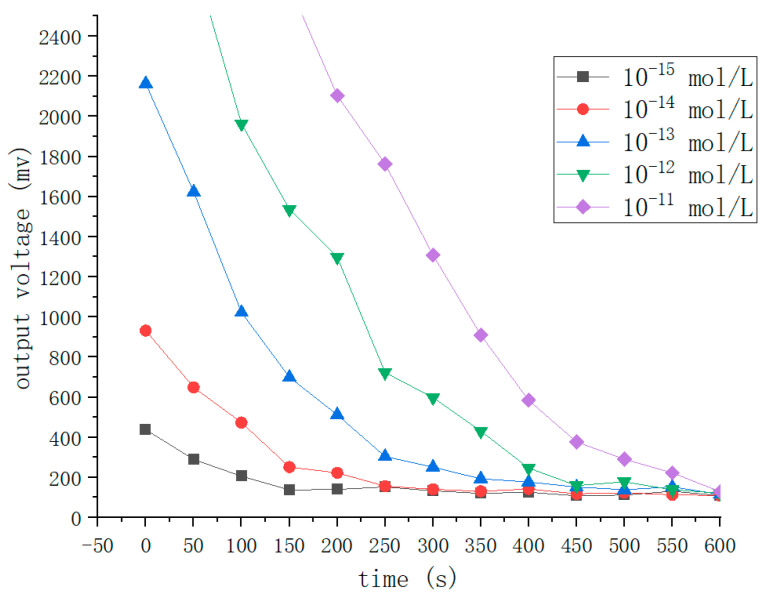
Kinetic fitting curves for the samples of five different ATP concentrations.

**Figure 10 nanomaterials-12-02417-f010:**
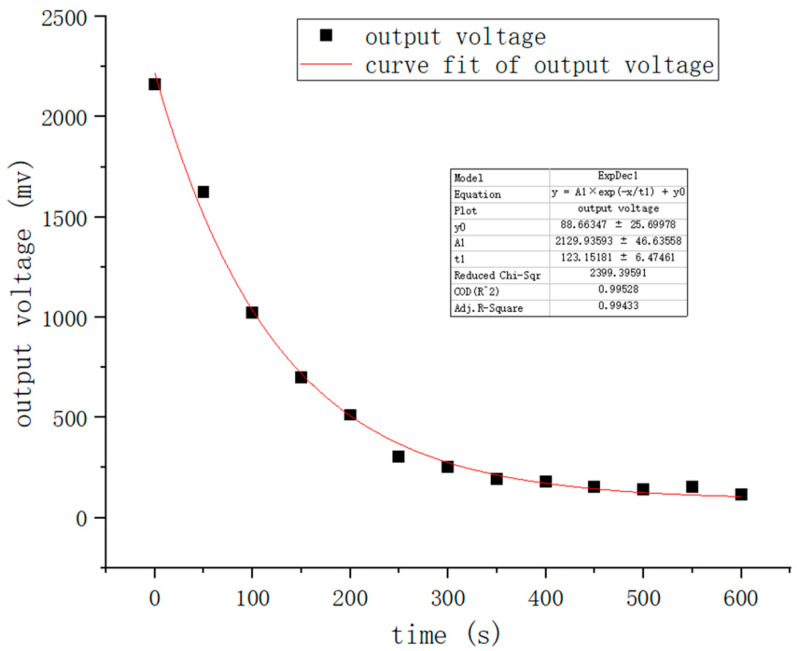
Kinetic fitted curve of the sample of concentration 10^−13^ mol/L.

**Figure 11 nanomaterials-12-02417-f011:**
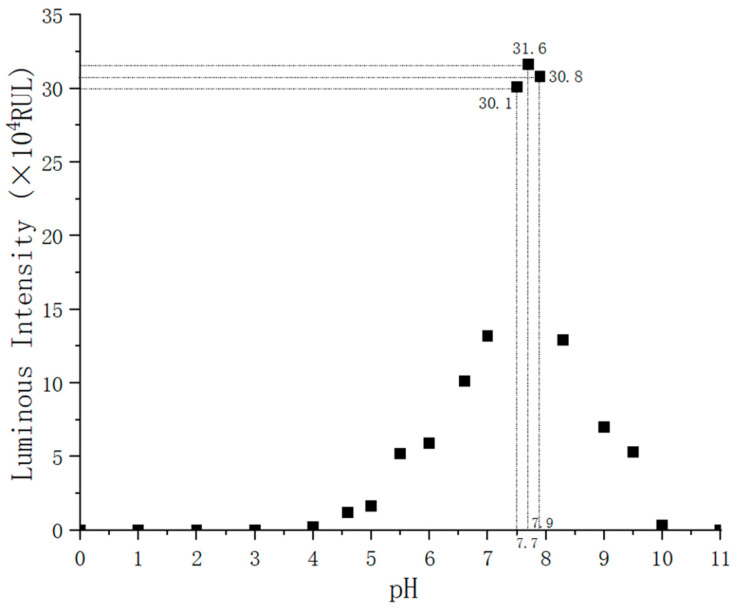
Effect of pH on luminescence intensity.

**Figure 12 nanomaterials-12-02417-f012:**
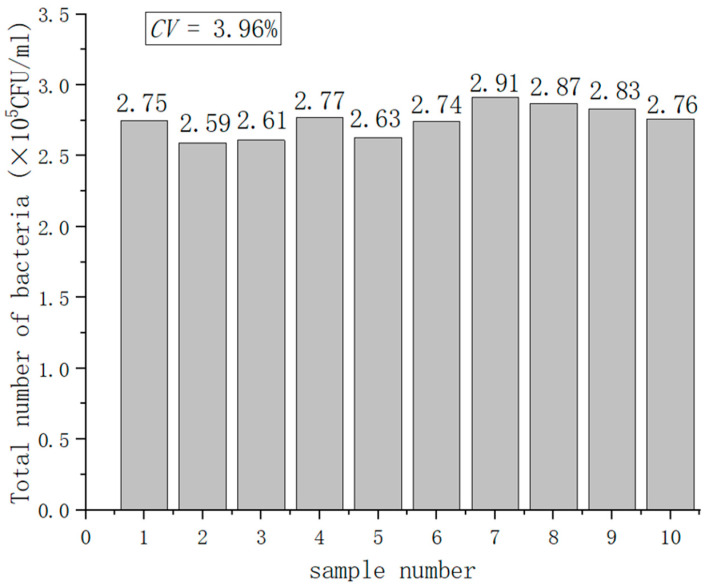
System precision testing.

**Figure 13 nanomaterials-12-02417-f013:**
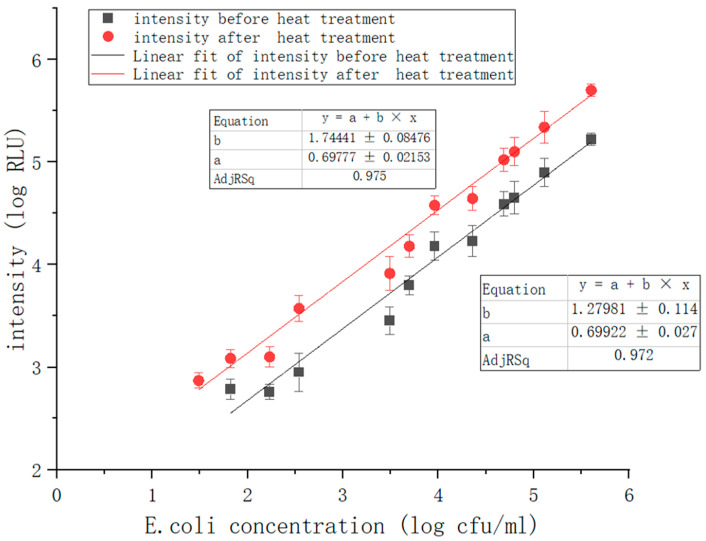
Correlation between luminescence intensity and *E. coli* concentration.

**Figure 14 nanomaterials-12-02417-f014:**
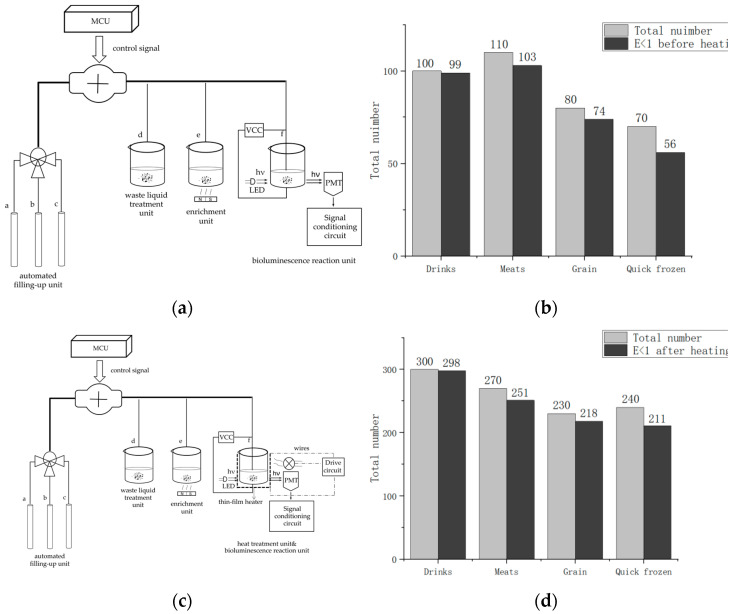
Comparison of the system accuracy. (**a**) detection model before adding the heat treatment module; (**b**) testing results before adding the heat treatment module; (**c**) detection model after adding the heat treatment module; (**d**) testing results after adding the heat treatment module.

**Table 1 nanomaterials-12-02417-t001:** The efficiency of binding *E. coli* antibodies to immunomagnetic beads.

A0	A1	R_C_
		%
34.2	4.6	86.5
35.8	5.4	84.9
33.9	4.8	85.8
37.4	5.1	86.4
38.2	4.9	87.2
36.5	4.7	87.1

**Table 2 nanomaterials-12-02417-t002:** Results compared with other test methods.

Method	Bacteria	Detection Range	Detection Time	Linear Coefficient	Reference
Electrochemical method	*E. coli K1:H7*	22 CFU/mL	30 min	0.841	[45]
Resistive method	*E. coli Rosetta 2pLyss*	10^7^ CFU/mL	2 h	0.875	[46]
Magnetic silica nanotubes method	*S. Typhimurium*	10^3^–10^7^ CFU/mL	30 min	0.901	[47]
Magnetic nanoprobe-ATP method	*E. coli O157:H7*	10^2^–10^8^ CFU/mL	<20 min	0.964	[19]
Nanoprobe transparent graphene electrode-ATP method	*E. coli O157:H7*	10^2^–10^6^ CFU/mL	<20 min	0.972	[20]
Nanoprobe- improve ATP method	*E. coli O157:H7*	31–10^6^ CFU/mL	25–30 min	0.975	Present work

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
