# Peer review of "A System of Rapidly Detecting Escherichia Coli in Food Based on a Nanoprobe and Improved ATP Bioluminescence Technology"

_nanomaterials, 2022, doi:10.3390/nano12142417_

Round 1
Reviewer 1 Report
The Manuscript ID: nanomaterials-1777517 by Sun et al. developed an Escherichia coli detection platform based on immunomagnetic separation in conjunction with ATP biofluorescence and graphene transparent electrodes. Validation experiments showed that this detection platform had detection range of 31 CFU/ml to 10^6 CFU/ml. The feasibility of using the Nanoprobe improved ATP biofluorescence method for E. coli detection was tested in various food matrices. The Authors described detailed information on the design and composition of the system; however, details on the culturing and preparation of the targeted bacterium were lacking. The manuscript also claimed that this method surpassed all current detection methods for E. coli; however, the manuscript lacked any information on how these experiments employing food matrices were performed.
L31-35: Please provide reference for statements on the incidence of bacterial infections.
L39 and L580-581: Not all E. coli are pathogenic. Please revise sentences. Only certain strain of E. coli harboring virulence factor are considered to be pathogenic to humans.
L52-60: Statements should be revised since the current methods listed are being used to meet industry needs. Please include immunoassays since these assays are being used by industry. Please include genome sequencing as a detection method.
L191-201: Please provide more information in this manuscript on the description of the antibody used for capturing Escherichia coli cells. What is the antigen detected by this antibody used in the present study?
L554-574: The Authors need to provide more information in the Materials and Methods section on the growth and culturing conditions for the E. coli strains used in the present study. Also, the results shown in Figure 14 lack sufficient information on how these experiments in food matrices were performed.
Author Response
Response to Reviewer 1 Comments
Dear Reviewer,
Thank you for your comments and suggestions, and we will respond to you below. In our attached revised draft, the changes made to your comments and suggestions are in red.
Point 1: L31-35: Please provide reference for statements on the incidence of bacterial infections.
Response 1: We have added a reference in line 36 in the article that can support the statement on the incidence of bacterial infections. The reference is Scharff, R.L. Economic Burden from Health Losses Due to Foodborne Illness in the United States. J. Food Prot.2012, 75, 123-131.
Point 2: L39 and L580-581: Not all E. coli are pathogenic. Please revise sentences. Only certain strain of E. coli harboring virulence factor are considered to be pathogenic to humans.
Response 2: Thank you for your correction, and the previous statement is not strict.E. coliis a normal host bacterium in the intestine of animals, and only a small secction of the E. colistrainswithvirulence factor can cause disease under certain conditions. So, E. coliis a conditionally pathogenic bacterium.We have revised in lines 39-42 and 642-644 in the article.
Point 3: L52-60: Statements should be revised since the current methods listed are being used to meet industry needs. Please include immunoassays since these assays are being used by industry. Please include genome sequencing as a detection method.
Response 3: Thank you for your correction, here our previous expression was ambiguous and not rigorous. We have revised in lines 55 - 68 of the article to include genome sequencing as a new modern assay. We have also used immunoassays as an example to show that the vast majority of the new modern methods are meeting industry needs. And somedrawbacks of thesenew modern methods are also illustrated. At the same time, the disadvantages of traditional methods are explained and it is argued that not all traditional methods can meet the industrial needs in every situation.
Point 4: L191-201: Please provide more information in this manuscript on the description of the antibody used for capturing Escherichia coli cells. What is the antigen detected by this antibody used in the present study?
Response 4: Antibodies are an immunoglobulin produced by plasma cells differentiated from B cells in response to the stimulation of antigenic substances, and antibodies can specifically bind to the corresponding antigens. In the article, our subject isE. coli O157:H7, so we choose to use E. coli O157:H7 monoclonal antibodies to capture them.E. coli O157:H7 has two antigens, the somatic antigen (O antigen) and the flagellar antigen (H antigen). The monoclonal antibodies bind specifically to the corresponding antigens, respectively, to achieve the effect of capturing the bacteria.Wehave revised in lines 217-223 in the article.
Point 5: L554-574: The Authors need to provide more information in the Materials and Methods section on the growth and culturing conditions for the E. coli strains used in the present study. Also, the results shown in Figure 14 lack sufficient information on how these experiments in food matrices were performed.
Response 5: We have added information on the growth and culture conditions of E. coli O157:H7 in lines 289-296 of the article. The process is to obtain a high concentration of the original bacterial solution using bacterial enrichment culture, and then the original bacterial solution is diluted to obtain the desired concentration of the sample solution. At the same time, we have complemented Figure 14 by adding the food detection prototype. And we have added the preparation process of food detection samples and their testing conditions in lines 615-624 of the article.
We deeply appreciate your consideration of our manuscript, and we look forward to receiving comments from you.
Thank you very much for your attention.
With kind regards
Your sincerely,
Zhen Sun, Jia Guo, Wenbo Wan, Chunxing Wang
Author Response
Response to Reviewer 2 Comments
Dear Reviewer,
Thank you for your comments and suggestions, and we will respond to you below. In our attached revised draft, the changes made to your comments and suggestions are in blue.
Point 1: The authors throughout the paper use the term "fluorescence" instead of "luminescence" or "bioluminescence". This is a gross error. There is no term "ATP fluorescence". Authors should make corrections throughout the all paper both in the text and in Figures 3, 5, 7.
Response 1: Thank you for your correction, and the previous statement is not strict.Wehave revised improper wording ofthe article. For example, in line 124, “the ATP fluorescence detection”has been changed to “the ATP bioluminescence detection”.Andthen wehave modified Figures 3, 5, and 7in lines 269, 391 and 434.
Point 2: Line 174 . Substrate of luciferase is luciferin, but not “Fluoresce”. Replace "Fluoresce" with "luciferin".
Response 2: We have modified equation (1) in line 188 of the article by replacing "Fluoresce" with "Luciferin".
Point 3: In Fig. 10 sidebar should be in English, not in Chinese
Response 3: Thank you for your correction, and this is a stupid mistake.We have revised Figure 10 in line 517 of the article.
Point 4: In Fig.11, It is necessary to write pH, not PH
Response 4: Thank you for your correction, and the previous figure is not strict.We have revised Figure 11 in line 542in the article.
Point 5: It is necessary to specify the composition of lysis reagent and bioluminescent reagent
Response 5: The composition of lysis reagent is 50 mmol/L Tris-HCl, 150 mmol/L NaCl, 0.02%NaN3, 100μg/mL PMSF, 1μg/mL Apro-tinin and 1% Triton X-100.The composition ofbioluminescent reagent is luciferase/luciferin regent substrate,which from Beijing YPH Biotechnology Co, LTD. We have added these information in lines 384-387of the article.
We deeply appreciate your consideration of our manuscript, and we look forward to receiving comments from you.
Thank you very much for your attention.
With kind regards
Your sincerely,
Zhen Sun, Jia Guo, Wenbo Wan, Chunxing Wang
Round 2
Reviewer 1 Report
The Authors have address all comments. This Reviewer endorses the publication of the manuscript.
This manuscript is a resubmission of an earlier submission. The following is a list of the peer review reports and author responses from that submission.